# Superglassy Polymers to Treat Natural Gas by Hybrid Membrane/Amine Processes: Can Fillers Help?

**DOI:** 10.3390/membranes10120413

**Published:** 2020-12-10

**Authors:** Ahmed W. Ameen, Peter M. Budd, Patricia Gorgojo

**Affiliations:** 1Department of Chemical Engineering and Analytical Science, School of Engineering, The University of Manchester, Oxford Road, Manchester M13 9PL, UK; ahmed.ameen@postgrad.manchester.ac.uk; 2Research & Development Center, Saudi Aramco, Dhahran 31311, Saudi Arabia; 3Department of Chemistry, School of Natural Sciences, The University of Manchester, Oxford Road, Manchester M13 9PL, UK; peter.budd@manchester.ac.uk

**Keywords:** gas separation, mixed matrix membranes (MMMs), superglassy polymers, PIM-1, natural gas, hybrid membrane/amine process

## Abstract

Superglassy polymers have emerged as potential membrane materials for several gas separation applications, including acid gas removal from natural gas. Despite the superior performance shown at laboratory scale, their use at industrial scale is hampered by their large drop in gas permeability over time due to physical aging. Several strategies are proposed in the literature to prevent loss of performance, the incorporation of fillers being a successful approach. In this work, we provide a comprehensive economic study on the application of superglassy membranes in a hybrid membrane/amine process for natural gas sweetening. The hybrid process is compared with the more traditional stand-alone amine-absorption technique for a range of membrane gas separation properties (CO_2_ permeance and CO_2_/CH_4_ selectivity), and recommendations for long-term membrane performance are made. These recommendations can drive future research on producing mixed matrix membranes (MMMs) of superglassy polymers with anti-aging properties (i.e., target permeance and selectivity is maintained over time), as thin film nanocomposite membranes (TFNs). For the selected natural gas composition of 28% of acid gas content (8% CO_2_ and 20% H_2_S), we have found that a CO_2_ permeance of 200 GPU and a CO_2_/CH_4_ selectivity of 16 is an optimal target.

## 1. Introduction

Generating power from natural gas to meet global demand involves lower capital expenses, lower carbon emissions, and higher thermal efficiencies as compared to other fossil fuels [1]. Raw natural gas contains mainly methane (CH_4_), typically 50% to 90%, with other hydrocarbons such as ethane (C_2_H_6_), propane (C_3_H_8_), butane (C_4_H_10_), and traces of heavier hydrocarbons. In addition, natural gas might contain undesirable impurities at varying compositions depending on the reservoir location, including carbon dioxide (CO_2_), water (H_2_O), nitrogen (N_2_), hydrogen sulfide (H_2_S), helium (He), etc. Among these impurities, CO_2_ and H_2_S (acid gas) need to be separated and removed from natural gas to prevent corrosion in the pipelines. According to US pipelines specification [2], CO_2_ content in natural gas should not exceed 2 mol%, while H_2_S should be less than 4 ppm. Moreover, more extensive removal of CO_2_ (<50 ppm) is essential when it is planned to transport the gas as a liquid (LNG), to prevent the freezing of CO_2_ [3].

The U.S. Energy Information Administration (EIA) in their International Energy Outlook report for 2019 (IEO2019) [4] predicted that the natural gas global consumption would increase by around 44%, from approximately 133.1 trillion cubic feet (Tcf) in 2018, to reach 191.4 Tcf by 2050. This expected increase in consumption, is leading natural gas producers to start to extract lower quality gas that contains higher amounts of impurities. It is well-documented that the Middle East has large natural gas reservoirs that contain around 30% of H_2_S and 10% of CO_2_, and Russia’s Astrakhan field contains 40% acid gases (around 25% H_2_S and 15% CO_2_) [5].

Currently, the most widespread method of removing CO_2_ from natural gas is the use of alkanolamine solvents in an absorption unit. To be more precise, absorption accounts for 70% of the technologies applied to remove CO_2_ from natural gas [6]. The amine-based absorption process is illustrated in Figure 1 [7]. Sour gas is first introduced at the bottom of the absorber column at high pressure, while lean amine (amine with low acid gas content) is introduced from the top of the column. The lean amine absorbs the acid gas present in the sour gas as it goes down the column and the treated sour gas (sweet gas) leaves from the top of the unit with very low acid gas content. The solvent, now with high acid content, leaves from the bottom and is subsequently sent to the regenerating unit. In this column, acid gas is removed from the amines at high temperature and low pressure before being recycled back to the absorber. If the acid gas stream has a high amount of H_2_S (above 20%), it can be further treated in the sulfur recovery unit (SRU) to convert H_2_S to sulfur.

However, the amine stand-alone process suffers from high operating and maintenance costs; the high operating cost is mainly related to the heat duty requirement to strip the acid gas from the amines in the regenerator. To ensure the best performance, the heating and cooling steps for the amine-based solvent need to be carefully monitored, as well as the degradation of amines to avoid equipment failure [8].

Among various potential alternative technologies to remove acid gases from natural gas, membrane technology stands out due to its simplicity and low operating cost. Since the installation of the first membrane unit using CO_2_-selective cellulose acetate (CA) membranes in 1981 [6,9], their use has risen, especially for low volume size applications below 30 million standard cubic feet per day (MMSCFD) [9,10,11]. However, membrane-based processes can also compete with the stand-alone amine process at high gas volume size when the CO_2_ concentration in the feed stream is higher than 10% [11].

Despite the above-mentioned advantages, membranes struggle to remove acid gas to ppm levels when H_2_S is present in the natural gas. In addition, membrane-based separation leads to higher loss of hydrocarbons. Recently, it has been proposed combining both processes for high flow rate and high acid gas content, where a membrane unit is installed before introducing the sour gas to the amine process (see Figure 2). In this system, the membrane unit removes a portion of the acid gas from the feed raw gas, consequently, feeding the amine unit with lower acid gas content. This will result in reducing amine capital and operating costs.

The membrane performance is a critical factor for the feasibility of the hybrid membrane/amine process. Only a few materials are commercially used for CO_2_ removal applications, all glassy polymers [8,12,13] due to their higher CO_2_ selectivity as compared to rubbery membranes.

A new type of glassy polymer membrane with high free volume, named superglassy polymer membranes, has emerged as a potential replacement for the conventional glassy membranes due to its high CO_2_ permeability, with moderate gas selectivities [14,15,16]. However, the main drawback of these superglassy membranes is the drop in performance over time, which is attributed to physical aging [17,18]. The most common types of superglassy membranes are polymers of intrinsic microporosity (e.g., PIM-1) and polyacetylenes (e.g., poly(1-trimethylsilyl-1-propyne [PTMSP] and poly(4-methyl-2-pentyne) [PMP]). They all suffer from a severe drop in permeability over time [18], which becomes even more pronounced for thin films [19].

Mixing superglassy polymers with porous and non-porous fillers has proven an attractive route to overcome the physical aging. Freestanding mixed matrix membranes (MMMs) have been successfully fabricated with metal organic framework (MOFs) [20,21], zeolites [22], silica [23], graphene-based materials [24,25], porous aromatic frameworks (PAFs) [26,27], and hypercrosslinked polymers (HCPs) [28,29,30,31]. The feasibility of using superglassy polymers in the hybrid membrane/amine process mainly depends on the long-term performance of the polymers under industrial operating condition, when produced as thin films to maximize CO_2_ permeance.

Some recent work on thin superglassy membranes for gas separation applications is summarized in Table 1. Liu et al. [32] recently tested thin-film composite (TFC) membranes based on PIM-1 as the selective layer on a polyacrylonitrile (PAN) support with a polydimethylsiloxane (PDMS) gutter layer. PIM-1 was mixed with two different types of MOFs (MOF-74-Ni and NH_2_-UiO-66) and pure gas testing was performed for CO_2_ and N_2_ gas, and compared with neat PIM-1 after 8 weeks. CO_2_ permeances of neat PIM-1, MOF-74-Ni/PIM-1, and NH_2_-UiO-66/PIM-1 TFC membranes were 490, 1200, and 900 GPU, respectively, and calculated ideal CO_2_/N_2_ selectivities 31, 30, and 26, respectively. Although CH_4_ was not included in their work, CH_4_ permeance is expected to be around 25% higher than N_2_. Although the results look promising, it is worth mentioning that the membrane performance is expected to drop when tested at higher pressures with gas containing hydrocarbons and acid gases due to plasticization phenomena.

Furthermore, membrane performance can be significantly altered in the presence of H_2_S, resulting in lower CO_2_/CH_4_ selectivity. Compared to CO_2_ (kinetic diameter = 3.3 Å), H_2_S (kinetic diameter = 3.6 Å) has lower diffusion coefficient. In normal glassy membranes, CO_2_ tends to permeate faster than H_2_S as the diffusion selectivity dominates the separation, leading to relatively low H_2_S/CH_4_ separation and thus hindering the use of glassy membranes to treat natural gas with high H_2_S concentrations [40]. However, the glassy membrane performance can be altered when applied at high acid gas partial pressures. For example, the H_2_S permeance might slightly exceed the CO_2_ permeance due to plasticization and sorption site competition [41,42,43].

In superglassy membranes, H_2_S permeates faster than CO_2_ due to a higher solubility factor, as recently reported by the research groups of William Koros and Ingo Pinnau [44], who tested dense PIM-1 and amidoxime-functionalized PIM-1 (AO-PIM-1) membranes with high-pressure ternary mixed feed gas (20% H_2_S, 20% CO_2_, and 60% CH_4_). To the best of our knowledge, this is the only work reporting superglassy membranes tested with gas mixtures close to real natural gas condition. In their work, an excellent membrane performance (H_2_S/CH_4_ = 30, H_2_S/CO_2_ = 4 for neat PIM-1 and H_2_S/CH_4_ = 75, H_2_S/CO_2_ = 5.5 for AO-PIM-1) was achieved even at a pressures above 1000 psi. The presence of high H_2_S content, which has much more effect on membrane plasticization than CO_2_, led to an exceptional H_2_S/CH_4_ selectivity at high-pressure feed. However, they only tested 50–60 µm thick membranes and no higher hydrocarbons were added to the feed mixture. In another paper, Merkel and Toy [40] found that H_2_S/CO_2_ selectivity in PTMSP is 1.2 at low-pressure feed gas mixture (1.5% H_2_S, 10.5% CO_2_, 46% CO, and 42% H_2_).

Several techno-economic studies have been carried out to study the feasibility of the hybrid membrane/amine process at different feed conditions. Bhide et al. [45] and Rezakazemi et al. [46] studied the effect of combining a commercial cellulose acetate (CA) membrane (CO_2_ permeance ~90 GPU, and CO_2_/CH_4_ selectivity ~21) with a diethanolamine absorption system at a feed flow rate of 35 and 50 MMSCFD respectively with a natural gas stream composition of 25% CO_2_ and trace amounts of H_2_S. In these two studies, the feed pressure was 800 psi, with an estimated membrane cost of 108 $/m^2^. It is worth noting that at field conditions, the selectivity value of the CA membrane is expected to be less than the assumed value. Despite the hybrid membrane/amine process leading to higher hydrocarbon loss, results by Bhide et al. [45] showed that the hybrid membrane/amine process becomes more feasible than the amine stand-alone process for CO_2_ content in the feed > 25%.

Hamad et al. [47] investigated the feasibility of one-stage and two-stage hybrid membrane/amine processes using two sets of performance data: one replicating the CO_2_ permeance and CO_2_/CH_4_ selectivity of a commercial membrane (50 GPU and 15, respectively) and another one with values that were double those of the first set. The simulated feed stream contained CH_4_ with no heavier hydrocarbons, 10% CO_2_ and 20% H_2_S, which made it possible to send the permeate gas exiting the membrane unit to the SRU, where more steam was produced in the waste heat boiler due to hydrocarbon (CH_4_) losses.

In their study, they found that the hybrid one-stage membrane/amine process was more feasible than the hybrid one with a two-stage membrane system despite the reduction in hydrocarbon loss in the two-stage process. In addition, the hybrid process with the commercial membrane was found to be less attractive than the amine stand-alone process. However, doubling the CO_2_ permeance (100 GPU) and the CO_2_/CH_4_ selectivity (30) led to a more competitive process.

In our work, an economic study is conducted for a hybrid one-stage membrane/amine process (hereafter referred to as “hybrid process”) and compared to a stand-alone amine process at a typical natural gas feed composition. The membrane unit assumed containing a superglassy membrane with a CO_2_ permeance in the range 50–800 GPU and a CO_2_/CH_4_ selectivity between 6 and 32, at three different feed pressures: 900, 600, and 300 psi. The permeate pressure is kept at 30 psi so the pressure ratio values (φ) are 30, 20, and 10, respectively. Recommendations for the required performance of the polymer leading to economic feasibility are provided, so that future research is geared towards the production of MMMs with target long-term CO_2_ permeance and CO_2_/CH_4_ selectivity.

Similar to what happens in rubbery membranes, heavier hydrocarbons permeate faster than lighter ones in superglassy membranes [48]. This is an important fact that deserves special attention when the concentration of heavier hydrocarbons in the raw gas is high, since high hydrocarbon losses can make the hybrid process unfeasible. In this study, raw natural gas containing 5 mol% ethane and 0.9 mol% propane is assumed.

Another important aspect to consider for the simulation and economic evaluation of the hybrid process is the feed flow rate. In our study we have selected a natural gas flow rate of 100 MMSCFD with an acid content of 28% (8% CO_2_ and 20% H_2_S). At this flow rate and composition, the stand-alone amine process can treat the gas with only one amine-absorption unit, which will allow a reasonable comparison between the two processes.

In general, for the sweetening process, several factors play an important role in studying the feasibility of a hybrid process. These factors are:acid gas permeanceCO_2_/CH_4_ and H_2_S/CH_4_ selectivitypressure ratio between feed and permeate sidefeed gas flow rateacid gas content in the feedacid gas removal fraction in the membrane processhydrocarbon content (C_2_+) in the feed gasmembrane process design, andmembrane module design

Other factors also need to be considered, such as raw gas extracting cost, sale gas cost, membrane skid cost, and membrane replacement rate. It is essential to study the feasibility of the hybrid process separately for each specific case. In this work, the effect of most of these factors is considered. After setting the superglassy membrane performance that leads to a more economically feasible process than the amine stand-alone process, the feed flow rate and acid gas content (fixed for the base case at 100 MMSCFD and 28%, respectively) are varied. This is done in order to assess the range where the hybrid process is still economically more viable. Finally, membrane cost and variations in ethane content in the feed are analyzed.

## 2. Case Study

### 2.1. Gas Condition 

The natural gas feed condition and composition assumed in this study is shown in Table 2. Our target is to remove the acid gas from hydrocarbons to meet pipeline specifications (CO_2_ < 2 mol% and H_2_S < 4 ppm). In the hybrid process, the membrane unit first reduces the CO_2_ and H_2_S content in the feed, and the amine-absorption treatment afterwards further reduces the remaining amount until the target values. An acid gas removal value of 60% in the membrane unit was found to be the most profitable target for a membrane with a CO_2_ permeance of 200 GPU and a CO_2_/CH_4_ selectivity of 16, and this value was fixed for all the simulations, as it had little effect on the selected economic indicators. This study also assumes that compression is not required to increase the pressure of the raw gas. Natural gas can reach the amine plant at different pressures depending on many factors such as natural gas resources, distance of amine plant from the gas reservoir, etc.

### 2.2. Membrane Performance

Gases permeate though non-porous membranes based on a solution-diffusion mechanism. The diffusion coefficient is mainly determined by the kinetic diameter of the gases, while the solubility coefficient is mainly determined by their critical temperature. Based on the above, the gases with highest to lowest diffusion coefficient should follow this order: H_2_O (2.65 Å) > CO_2_ (3.3 Å) > H_2_S (3.6 Å) > N_2_ (3.64 Å) > CH_4_ (3.8 Å) > C_3_H_8_ (4.3 Å) [49]. On the other hand, solubility coefficient should follow the following order: H_2_O (647.3 K) > H_2_S (373.2 K) > C_3_H_8_ (369.8 K) > C_2_H_6_ (305.3 K) > CO_2_ (304.2 K) > CH_4_ (190.6 K) > N_2_ (126.2 K) [50].

In this work, membranes with a CO_2_ permeance in the range 50–800 GPU and CO_2_/CH_4_ selectivities of 6, 10, 16 and 32, for each of the studied CO_2_ permeances, are evaluated. The H_2_S/CO_2_ selectivity is fixed at 1.2 in all cases.

Heavier hydrocarbons are expected to permeate faster than methane. Thomas et al. [48] reported an ethane permeance 2.5 times higher than the methane one in dense PIM-1. Therefore, in this study, the C_2_H_6_/CH_4_ selectivity is assumed at 2.5 while the C_3_H_8_/CH_4_ selectivity has been set at 6.25 (propane permeance 2.5 times higher than ethane permeance). Moreover, the N_2_ permeance is assumed to be 25% lower than the CH_4_ one.

### 2.3. Amine Solvent Types and Selection

Amines are stable colorless liquids widely used in acid gas removal processes due to their low relative cost. In addition, most amine types can be processed at high temperature up to their boiling points without decomposition. Several types of amine, such as monoethanolamine (MEA), diethanolamine (DEA), diisopropanolamine (DIPA), diglycolamine (DGA), and methyldiethanolamine (MDEA), are widely used in this application [1]. In this study, DGA is chosen as the amine-based solvent for the following reasons:the capability of removing carbonyl sulfide (COS) and mercaptans from gas beside H_2_S and CO_2_:the DGA concentration in solution is usually 50 to 60% higher than other amines;a higher amine concentration means higher acid gas uptake, lower circulation rate, and lower electric power and heating duty requirements [1]; andthe corrosion in the process using DGA solvent is slightly less than when using the MEA [7].

However, other aspects such as solvent degradation and solubility of heavier hydrocarbons need to be carefully considered when selecting the amine. It is recommended to use commercial software such as Aspen Plus to compare the performance of different solvents, as in the work by Bae et al. [51], where it was found that DGA led to significant energy savings as compared to MEA for natural gas containing high CO_2_ concentration.

### 2.4. Total Capital Investment (TCI) 

The total capital investment (TCI) is the sum of the fixed capital investment (FCI) and the working capital investment (WCI). How to calculate these two economic indicators is explained in the following sections.

#### 2.4.1. Amine Process Capital Cost

The capital cost for the amine process is estimated from the paper by Hamad et al. [47]. The main equipment for the amine plant is the absorber, the regenerator, the heat exchangers, pumps, air cooler fans, and a surge tank. The cost of each piece of equipment is estimated from Equation (1):(1)$$C_{A}=C_{B}{\left(\frac{S_{B}}{S_{A}}\right)}^{m}\times \left(\frac{I_{A}}{I_{B}}\right)\times L_{f}$$
where:

$C_{A}$ is the equipment cost in this study

$C_{B}$ is the equipment cost calculated by Hamad et al. [47]

$\frac{S_{B}}{S_{A}}$ is the equipment capacity ratio between that in the work by Hamad et al. [47] and this study

$I_{A}$ and $I_{B}$ are the Chemical Engineering Plant Cost Index (CEPCI) in 2019 and 2017, respectively.

$L_{f}$ is the estimated location factor for the Middle East (=1.07) [52]

In addition, the equipment costs are related to the content of acid gas in the feed. The cost exponent factor m is different for the different units, and is estimated based on the information provided by A.K. Coker’s book [53,54]; m values are shown in Table 3.

#### 2.4.2. Membrane Unit Capital Cost

The main cost of the membrane unit is the skid as it represents around 95% of the total cost. The membrane skid cost is not available directly as the suppliers usually offer complete whole packages, rather than individual membrane module cost. However, the membrane skid cost is estimated to be around 500 $/m^2^ for high-pressure applications [8].

The required membrane area is calculated by ASPEN HYSYS using the Saudi Aramco proprietary Industrial Membrane Process Simulator (IMPS). Single-stage spiral wound cross-flow is chosen as a type of membrane configuration. The membrane area obtained in the simulation is further increased by 15% to account for design margin and being able to replace part of the used membranes while the plant is operating.

#### 2.4.3. Fixed Capital Investment (FCI)

Direct and indirect cost factors and its factor to purchased equipment are listed in Table 4 [55]. Both the amine stand-alone process and the hybrid system are assumed to have the same factors, although lower real values are expected for the hybrid process due to its simplicity as compared to the stand-alone amine process. The total factor is the sum of the delivered cost factor, direct and indirect cost factors.

#### 2.4.4. Working Capital Investment (WCI)

In this work, the working capital cost is assumed to be the operating expenses for the first year in addition to the amine loading and start-up expenses. How the operating cost has been calculated is shown in the next section. The cost of the amine loading is estimated from the value reported by Hamad et al. [47] assuming it is proportional to the acid gas flow rate entering the amine unit. The start-up expenses are estimated at 10% of the fixed capital investment as suggested elsewhere [56].

### 2.5. Operating Expenses

#### 2.5.1. Membrane Operating Cost

Since the membrane is only operated as one stage and the feed is already at high pressure, membrane replacement is the only operating cost that needs to be considered for the membrane process. The membrane element price itself depends on the type of membranes and the configuration of the membrane. In addition, element prices have considerably decreased over the years, from around 100 $/m^2^ on average in 1990 to less than 50 $/m^2^ on average in 2015 [57,58]. The latter value has been chosen for this case study. In addition, membranes are assumed to be replaced twice a year.

#### 2.5.2. Amine Operating Cost

The amine operating costs comprise the cost of the amine make-up stream in the process, the cost of utilities for heating and cooling the amine system and electricity consumed by the pumps. All these costs are calculated based on the DGA circulation rate, which is directly proportional to the flowrate of the sour gas stream fed to the process *Q* (in MMSCFD) and the acid gas content (*y*), and inversely proportional to the concentration of DGA in the amine solution (*x*), as per Equation (2) [59,60]. This equation is valid when DGA is used as the amine-based solvent, assuming 0.39 mol acid gas pick-up per mole DGA.
(2)$$Amine\;circulation\;rate\left[GPM\right]=55.8\times \left(\frac{Q\times y}{x}\right)$$

In the sweetening gas treatment, some DGA can be lost due to degradation, vaporization, mechanical losses, and shutdown/start-up. Therefore, the amine content needs to be measured periodically to determine the amount of amine that needs to be added (amine make-up) to overcome the losses. The amine make-up (∀) is proportional to the sour gas flow rate (Q), and in this work is assumed to be 3 lb/MMSCF of sour gas flow. In addition, the DGA amine cost ($C_{DGA}$) is estimated at 1.85 $/lb. Plant working days (WD) is assumed to be 343 days per annum. Thus, annual amine make-up cost is calculated using Equation (3).
(3)$$amine\;makeup\;cost\left[\frac{\$}{yr}\right]=\forall \left[\frac{lb}{MMSCF}\right]\times C_{DGA}\left[\frac{\$}{lb}\right]\times WD\left[\frac{day}{yr}\right]\times Q\left[MMSCFD\right]$$

In addition, the required heat duty in the regenerator is directly proportional to the amine circulation rate. The total heating duty requirement for the amine unit mainly arises from the reboiler, rich/lean amine heat exchanger, amine cooler, and the reflux condenser (Figure 2). Table 5 shows the factor used to calculate the heating duty for each of these items of equipment.

The duty requirement needs to be considered carefully as it covers most of the operating costs of the amine process. It is worth mentioning that heat integration allows savings in the process as the majority of the required duty is provided by the waste heat boiler (WHB) from the SRU, which is massively affected by the content of hydrocarbons entering the WHB unit. ASPEN-HYSYS (SULSIM package) is used to estimate the duty generated by the WHB unit. In addition, the duty of the air blowers in the SRU unit is considered in this balance. The efficiency of WHB and air-blower are assumed to be 30% and 70%, respectively. 

Moreover, electric power needed for pumps and fans is calculated. Most of the electrical power required for the amine unit is consumed by the main amine pump, the amine booster pumps, and reflux pumps. The power required for these units is calculated using the factors listed in Table 6 [60].

#### 2.5.3. Other General Operating Expenses

In addition to the operating costs for the membrane and the amine process that have just been described, there are other expenses such as those associated with maintenance and repairs, estimated at 7% of the fixed capital cost, and operating supplies. Operating supplies are supplies that cannot be considered as a raw material or maintenance, such as lubricants or test chemicals. Their cost is estimated as 15% of the maintenance and repairs expenses [55].

Furthermore, worker wages are calculated by estimating the number of workers needed per shift and the average hourly wage salary of these workers. For this study, the number of workers needed per shift (ϖ) is assumed to be 10, while the average workers’ salary is 35 $/h. Hence, the annual operating workers’ wages can be calculated using Equation (4):(4)$$annual\;woker\;wages\;\left[\frac{\$}{yr}\right]=\varpi \times wokers\;salary\;\left[\frac{\$}{h}\right]\times 24\;\left[\frac{h}{day}\right]\times 365\;\left[\frac{day}{yr}\right]$$

An operations manager is also needed, and the cost is estimated to be 15% of the operating labor cost.

Finally, laboratory charges are added to the cost to account for laboratory tests and product quality control, which are estimated to be 10–20% of the operating worker’s wages [55]. The percentage value in this study is 15%.

### 2.6. Cash Flow and Revenue

In the gas sweetening process, the main product is the sweet gas leaving the absorber unit, while the acid gas stream is the by-product. Sweet gas departing from the amine unit is saturated with water. Water needs to be removed via a dehydration process to meet the sale gas specification. In some cases, higher hydrocarbons are recovered as they are more valuable than CH_4_. In this study, the dehydration process and recovery of higher hydrocarbons is not considered. The sale gas price varies widely depending on the natural gas market. In this study, the sale gas price is estimated at 2.3 $/MMBtu. The heating value (HV) of the sale gas in [Btu/SCF] is obtained from HYSYS software. Plant working days (WD) is assumed to be 343 days per annum Thus, the annual sale gas cost is estimated using Equation (5).
(5)Gas cost[$yr]=HV[BtuSCF]×Q[MMSCFD]×gas price[$MMBtu]×WD[daysyr]

On the other hand, the raw gas (feed gas) cost was not considered in the calculation of operating expenses, but is considered in the revenue calculations. Raw gas price depends on many factors, such as reservoir type, location, and depth. Raw gas price is estimated in this work at 1.2 $/MMBtu. Equation (5) can also be used to calculate the annual cost of raw gas.

Depreciation is applied to the FCI [61]. The linear method, which is the simplest and most commonly used, has been selected to calculate the depreciation. 

Finally, cash flow and net present value (NPV) are calculated over a 20-year period, while the construction of the plant is assumed to take three years. The inflation rate is assumed at 2.5%/year.

## 3. Results and Discussion

### 3.1. Membrane Area and Skid Cost

It is worth noting that the membrane unit is designed to remove 60% of the acid gas in all cases, with Aspen HYSYS determining the required membrane area. Figure 3 shows how the membrane area and cost are affected by the CO_2_ permeance of the membrane. For all pressure ratios (φ) and selectivity values, the membrane area curves follow a power function to CO_2_ permeance [$membrane\;area=a\times {\left(1+CO_{2}\;permeance\right)}^{b}$].

As seen in Figure 3, at a pressure ratio equal to 30, the curve turning point occurs when the CO_2_ permeance is between 100 and 200 GPU. The membrane area tends to increase to infinity as the CO_2_ permeance approaches a value of 0 GPU, with a very high membrane skid cost below a permeance of 100 GPU. It is also obvious from the graph in Figure 3 that increasing the permeance above 200 GPU only add a little advantage to the membrane process in terms of skid cost. Thus, at a pressure ratio of 30, the optimal CO_2_ permeance was found at around 200 GPU. When looking at a lower pressure ratio of 20, the optimal CO_2_ permeance range is slightly shifted to the right but it is close to the optimal value of 200 GPU. However, when the pressure ratio is further reduced down to 10, due to the lower driving force, the required membrane area for the studied permeance range is much larger, and thus the skid cost. It is difficult to give a critical value of permeance, as it is not as obvious as for the higher pressure ratios, but it could be around 400 GPU. On the other hand, increasing the pressure ratio above 30 will lead to a negligible reduction in membrane area, as the selectivity factors start to dominate the separation.

Moreover, the required membrane area does not change with the CO_2_/CH_4_ selectivity for high pressure ratios, and there is only a slight increase for a pressure ratio of 10. For a given CO_2_ permeance, higher CO_2_/CH_4_ selectivity leads to lower CH_4_ permeance and thus an increase in the composition of acid gas in the permeate side. This results in a reduction of the driving force for the acid gas through the membrane, leading to slightly lower permeation of acid gas and so higher membrane areas are required.

### 3.2. Total Capital Investment

The total capital investment (TCI) of the hybrid process is the sum of the amine and the membrane capital costs. TCI values for a range of hybrid processes using superglassy polymer membranes with different CO_2_/CH_4_ selectivities and CO_2_ permeance values are plotted in Figure 4 and compared to the TCI value of the amine stand-alone process (represented as a horizontal line in the graph). It is worth noting that the capital costs associated to the amine process for all the hybrid processes (not shown in the figure) are almost the same, since the membrane retentate stream (i.e., stream fed to the amine process) has the same acid gas content with a slight change in flow rate. Thus, the effect of CO_2_ permeance on the TCI of the hybrid process is similar to its effect on the membrane area and the skid cost discussed above.

At a pressure ratio of 30 and a CO_2_ permeance >100 GPU, the total capital cost of the hybrid processes is lower than that of the amine stand-alone one, with small variations for the studied CO_2_/CH_4_ selectivity values. Moreover, at a pressure ratio of 20 and CO_2_ permeance >200 GPU, the hybrid processes have lower capital cost, with still a negligible effect of CO_2_/CH_4_ selectivity. At a pressure ratio of 10, the CO_2_/CH_4_ selectivity has a more clear influence, but the TIC of the hybrid process only becomes more advantageous than the amine stand-alone process for membranes whose CO_2_ permeances are >400 GPU.

### 3.3. Total Annual Operating Cost

In general, hybrid processes have lower annual operating costs than the stand-alone amine absorption process, as seen in the graph in Figure 5, which is attributed to the lower heating duty requirement due to the lower amine circulation rate. In general, the higher the CO_2_ permeance of the membrane in a hybrid system, the lower the operating costs. This is mainly due to the low maintenance and replacement costs of the membranes; maintenance cost is estimated at 7% of the FCI, and membranes are typically replaced twice a year. Obviously, larger membrane areas lead to higher replacement cost.

The effect of CO_2_/CH_4_ selectivity on the operating costs is mainly driven by the balance of the heating duty consumed in the amine unit and the heating duty produced by the SRU unit. Lower CO_2_/CH_4_ selectivity leads to higher hydrocarbon losses, and therefore lower revenue (discussed in more detail in the next section). However, these losses can be utilized to produce more heat duty from the WHB in the SRU unit, which can be integrated within the amine unit, decreasing the operating costs associated with the use of utilities, as can be seen in Figure 5.

Regarding the pressure ratio, a larger value leads to lower operating costs, assuming long-term stability of the membranes. Similarly to the effect with CO_2_ permeance, a larger pressure ratio leads to lower FCI and, therefore, lower maintenance and replacement costs, as lower membrane areas are required.

### 3.4. Revenue and Hydrocarbon Losses

The effect of CO_2_/CH_4_ selectivity on revenues of the hybrid process at pressure ratios of 10, 20, and 30 is shown in Figure 6. Compared to the stand-alone amine process, the hybrid processes always have lower revenues due to higher hydrocarbon losses. In addition, the amine stand-alone process produces a larger amount of sweet gas with higher heating value due to higher content of hydrocarbons as compared to the hybrid processes.

In addition, when the hybrid process operates with a membrane with higher CO_2_/CH_4_ selectivity, the revenue value tends to increase, this due to the reduction of hydrocarbon losses. A similar effect can be seen at the three pressure ratios evaluated. The CO_2_ permeance does not have any influence on revenues, since membranes with the same CO_2_/CH_4_ selectivity and varied CO_2_ permeance produce an equal amount of permeate and retentate streams.

### 3.5. Net Present Value (NPV)

NPV values of the hybrid and the stand-alone amine processes are shown in Figure 7. For the hybrid process the NPV has been calculated at three different pressure ratios (10, 20 and 30) and four CO_2_/CH_4_ membrane selectivity values (6, 10, 16 and 32), and are plotted against the CO_2_ permeance in the range 50–800 GPU.

At a pressure ratio of 30 (feed pressure = 900 psi), the hybrid process appears to have a clear advantage over the stand-alone amine process for the studied range of CO_2_ permeance at CO_2_/CH_4_ selectivity of 16 and 32. However, CO_2_ permeances below 100 GPU dramatically reduce the feasibility of the hybrid process. It can be seen that for permeances above 200 GPU the effect on NPV values is not as pronounced as for lower CO_2_ permeances.

It is also worth noting that a membrane with a CO_2_/CH_4_ selectivity of at least 16 is preferred in order to maximize NPV, as the amount of hydrocarbon losses are reduced, and thus the sale gas production rate with higher heating value increased.

Generally, CO_2_ permeances have a clear effect on the total capital cost and total operating cost, while the CO_2_/CH_4_ selectivity has a bigger effect on revenues.

At a pressure ratio of 20 the hybrid process is still feasible at reasonable membrane performances. Lowering the pressure ratio slightly increases the hydrocarbon losses and so revenues are reduced. In addition, the capital and operating costs slightly increase. However, when the pressure ratio is further reduced to 10, the values of revenues, capital costs, and operating costs are all severely affected, with the stand-alone amine process more attractive even when the membrane in the hybrid process has a CO_2_ permeance of 400 GPU and a CO_2_/CH_4_ selectivity of around 25. It is also worth noting that such values of permeance are hard to be sustained in time in TFC membranes of superglassy polymers due to physical aging.

The feasibility of the hybrid process using a commercial CA membrane is also evaluated for the three pressure ratios. The CO_2_ permeance and CO_2_/CH_4_ selectivity of CA are assumed to be 100 GPU and 15, respectively based on field test results [62,63,64]. Unlike superglassy membranes, the CA membrane is expected to have lower H_2_S permeance than CO_2_, so a higher membrane area is required. On the other hand, a clear advantage of CA and all other glassy polymers is the low permeance of heavier hydrocarbons, which leads to lower hydrocarbons losses than those with superglassy membranes, and so increased revenues. Figure 7 shows that the hybrid process using CA can beat the stand-alone amine process at a pressure ratio of 30. This advantage is subject to the presence of acid gas in the feed above 25%. Lower acid gas content adds an additional challenge to the hybrid process using CA and other membranes in general. The effect of the acid gas content in the feed for superglassy membranes is discussed in more detail in Section 3.6.

The selectivity vs permeance graph in Figure 8 shows the membrane performance target that makes the hybrid process more profitable than the amine stand-alone process. Any points to the right-hand side of the pressure ratio lines indicate a hybrid process that outperforms that of the stand-alone amine.

At pressure ratios of 20 and 30, the hybrid process is economically feasible for reasonable membrane performance values. The minimum CO_2_ permeance required is 200 GPU with a CO_2_/CH_4_ selectivity > 15. Any performance exceeding these numbers will further improve the economic indicators of the hybrid process. Therefore, for pressure ratios in the range 20–30, a reasonable CO_2_ permeance of 200 and CO_2_/CH_4_ selectivity of 16 is set as the target for long-term performance of superglassy membranes. The patterned rectangle in Figure 8 indicates an acceptable range for membrane performance at pressure ratios above 20. The performance of a fresh hollow fiber PIM-1 membrane (coated in PDMS/PAN) [37] has been included in Figure 8. Another hollow fiber PIM-1 membrane performance (after an aging time of 2 months) is plotted in the graph [35]. Both would be suitable in a hybrid process provided they could maintain the performance under industrial gas conditions and composition for a period of 6 months. Some MMMs prepared as TFCs reported in the literature show promising results (see Table 1), although the performance for longer aging periods should be evaluated with membranes tested for real raw natural gas at field condition. However, there are other challenges that need to be considered alongside improving the anti-aging properties of TFCs such as reducing the selective layer thickness, optimizing the chemistry between the selective layer and the substrate and fabricating hollow fiber or spiral wound modules with minimal polymer consumption.

### 3.6. Effect of Feed Flowrate and Acid Gas Content 

The effect of acid gas content (15–32%) and feed flowrate (25–100 MMSCFD) is studied at a fixed pressure ratio of 20 considering three superglassy membrane performances:(i)CO_2_ permeance = 200 GPU & CO_2_/CH_4_ selectivity = 16 (target performance)(ii)CO_2_ permeance = 400 GPU & CO_2_/CH_4_ selectivity = 16 (doubled CO_2_ permeance)(iii)CO_2_ permeance = 200 GPU & CO_2_/CH_4_ selectivity = 32 (doubled CO_2_/CH_4_ selectivity)

It is worth noting that feed flowrates > 100 MMSCFD are not included in this study, due to the design limitation for the stand-alone amine process; feed flowrates higher than 100 MMSCFD with acid gas content of 20% or above usually requires two parallel amine units in the stand-alone process, while the hybrid process might only require one amine unit. The acid gas content has been varied by reducing the H_2_S content while keeping CO_2_ content fixed at 8%.

NPV values of the hybrid process at these performances and the amine stand-alone process are compared and summarized in Figure 9. The hybrid process’ preferred feed condition range is shown. At the set target performance, the hybrid process is more favorable when the acid gas content is above 21% at a feed flowrate of 100 MMSCFD. The reduction of feed flowrate slightly stretches the line to 20%. However, the hybrid process itself becomes not applicable when the acid gas content goes up (e.g., >32% at 100 MMSFD flowrate). This comparison has been performed assuming the sale gas price at $2.3/MMBtu. Increasing the sale price would widen this range to make the hybrid process more applicable at even higher acid gas content. 

As can be seen in Figure 9, doubling the membrane performance increases the preferred hybrid process working region at both ends. However, doubling the CO_2_/CH_4_ selectivity seems to be slightly more attractive than doubling the CO_2_ permeance. For example, for an acid gas concentration of slightly above 18%, doubling the CO_2_/CH_4_ selectivity makes the hybrid process more favorable than the stand-alone one (right hand side graph in Figure 9), whereas doubling the permeance is not enough (see middle graph in Figure 9). In general, increasing the selectivity means lower hydrocarbon loss and higher revenues, while increasing the permeance leads to less capital and operating cost. 

It is worth mentioning that Figure 9 represents a non-optimized process, but the optimization of the membrane unit could further widen the favorable range indicated in Figure 9. The optimization process can involve finding out the optimum percentage of acid gas removal, applying a two-stage process at very high feed acid gas content, or applying counter-current flow instead of crossflow. In addition, the feed pressure and sale gas price are important factors to consider, as discussed earlier. Also, a higher pressure ratio and sale gas price would widen the hybrid process applicable range.

### 3.7. Sensitivity Factors

In this section, some important factors are varied to understand their effect on the feasibility of the hybrid process over that of the amine stand-alone. The sensitivity studies were conducted for a membrane performance of CO_2_ permeance = 200 GPU and CO_2_/CH_4_ selectivity = 16, and a pressure ratio of 20. All other factors are fixed unless mentioned otherwise.

#### 3.7.1. Ethane Content in the Feed

The main disadvantage of superglassy membranes over normal glassy membranes is that heavier hydrocarbons (C_2_+) permeate faster than lighter ones. Thus, the presence of a high amount of ethane and propane will reduce the hybrid process feasibility when a superglassy membrane is used.

Figure 10 shows the effect of ethane content in the feed side. In all cases, the ethane content has been increased at the expense of reducing the amount of methane in the raw natural gas. As the ethane content in the natural gas increases, so does the NPV for both the hybrid and the amine stand-alone processes due to the higher heating value of ethane as compared to methane, although the gap between these two processes is reduced. Up to 20% of ethane, which is usually the maximum content found in raw gas, the hybrid process has a higher NPV Several approaches can be followed to reduce the effect of high hydrocarbon content in the feed, such as applying a two-stage membrane process with a superglassy membrane in the first unit and a normal glassy membrane in the second unit to recover the hydrocarbon losses. 

#### 3.7.2. Membrane Costs (Skid, Material, Replacement Rate)

Membrane skid cost is estimated at $500/m^2^ in this study. As the feed pressure goes down, the skid cost tends to decrease. Again, estimating the membrane price is quite challenging as most suppliers do not give the membrane unit price alone. Another factor is the membrane material cost and replacement rate. Membrane material cost is very low compared to skid cost. Thus, it has no major effect on the fixed capital cost of the process. However, operating costs can be slightly affected by these factors (material cost and replacement rate).

Thus, all three main membrane factors are varied in this section and the NPV values of the hybrid and the amine stand-alone processes are determined and compared. As seen in the graphs in Figure 11, by varying the membrane skid cost between $(200–700)/m^2^ the NPV value is hugely affected. However, even at the highest estimated price ($700/m^2^), the hybrid process is more advantageous than the amine stand-alone process.

Moreover, membrane material cost seems to have a minor effect on the hybrid process feasibility compared to the replacement rate. In the reference case, the membrane is assumed to be replaced twice a year. Increasing the replacing rate to 6 will make the hybrid process less feasible than the amine stand-alone. In addition to the reduction of the feasibility of the hybrid process at this rate, replacing all the membranes six times in one year looks to be completely not practical. However, Figure 11 shows clearly the need for membrane material than can have stable performance over a few months. Enhancing the superglassy membrane life to one year from six months will enhance the hybrid process by around 13%, adding a huge improvement to the hybrid process over the amine stand-alone.

As already discussed, one of the main drawbacks of superglassy membranes is the reduction of gas permeance over time. Reducing the aging of superglassy membranes can be achieved by mixing it with different kinds of organic and inorganic fillers. Some recent studies at laboratory scale show promising results in this area. However, applying superglassy membranes at industrial scale requires more research with membranes tested under real conditions, and pilot-scale units that confirm the suitability of the membranes for long-term operation.

## 4. Conclusions

In this work, a superglassy membrane performance target is set for the hybrid membrane/amine process to treat natural gas containing a relatively high amount of acid gas (8% CO_2_ and 20% H_2_S) at various feed pressures. At a CO_2_ permeance of 200 GPU and a CO_2_/CH_4_ selectivity of 16, tested at field conditions, superglassy membranes can be applied for the sweetening of natural gas. Generally, when the acid gas content is higher than 20%, the hybrid process becomes more feasible than that of the amine stand-alone at a feed pressure of 600 psi. Moreover, doubling the CO_2_/CH_4_ selectivity is more attractive than doubling the CO_2_ permeance. Researchers in the field working with superglassy polymers should take the recommended values in this work as a target performance to enhance the existing membranes.

In addition, it was found that the hybrid process is largely sensitive to the membrane skid cost and the membrane replacement rate, which highlights the importance of improving the membranes’ lifespan.

Several works report thin superglassy membranes that already meet the performance target, but with lack of long-term performance or lack of testing at field condition. It is expected that performance drops over time when tested at field condition. Thus, this opens an avenue for the use of more robust MMMs. In addition, other factors such as mechanical and thermal stability, and membrane plasticization resistance, are very important.

## Figures and Tables

**Figure 1 membranes-10-00413-f001:**
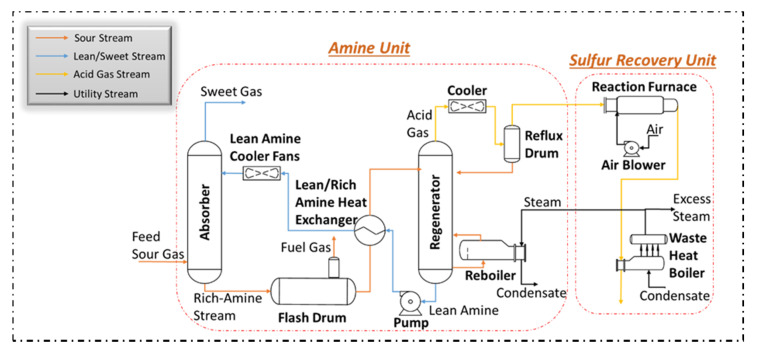
Simplified process flow diagram of the amine stand-alone process showing major equipment.

**Figure 2 membranes-10-00413-f002:**
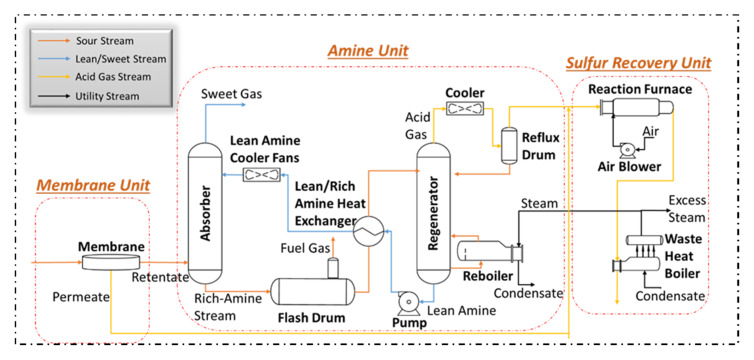
Simplified process flow diagram of the hybrid membrane/amine process.

**Figure 3 membranes-10-00413-f003:**
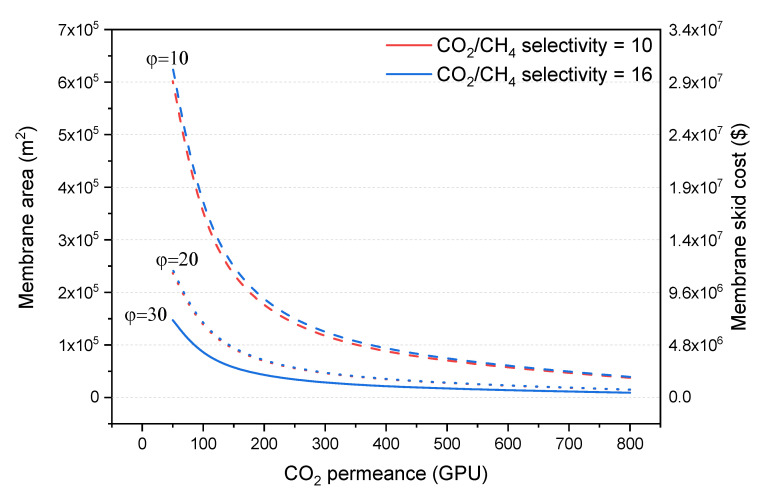
Effect of CO_2_ permeance on membrane area (left axis) and membrane skid cost (right axis) at CO_2_/CH_4_ selectivity values of 10 (red lines) and 16 (blue lines) and pressure ratio values (φ) of 10 (dash lines), 20 (dot lines) and 30 (solid lines).

**Figure 4 membranes-10-00413-f004:**
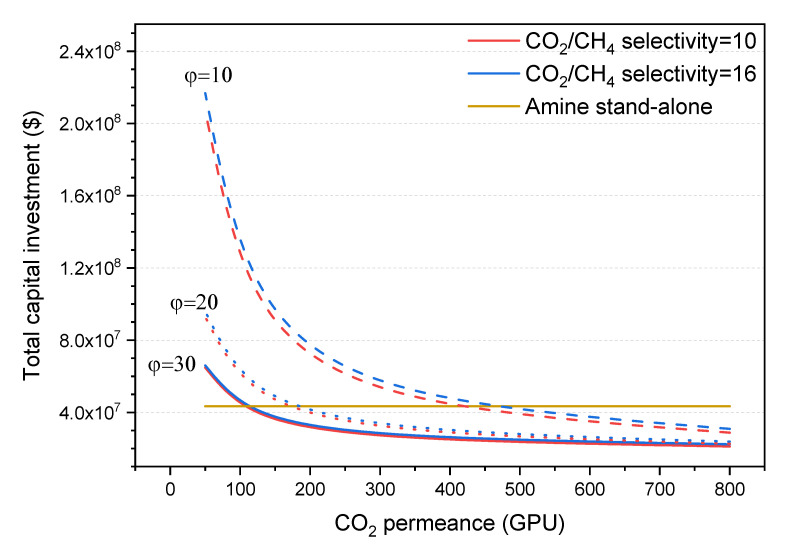
Effect of CO_2_ permeance on the TCI of the hybrid process at CO_2_/CH_4_ selectivity of 10 (red lines) and 16 (blue lines) and different pressure ratios of 10 (dash lines), 20 (dot lines), and 30 (solid lines). The cost is compared to the cost of the stand-alone amine process (horizontal dark yellow line).

**Figure 5 membranes-10-00413-f005:**
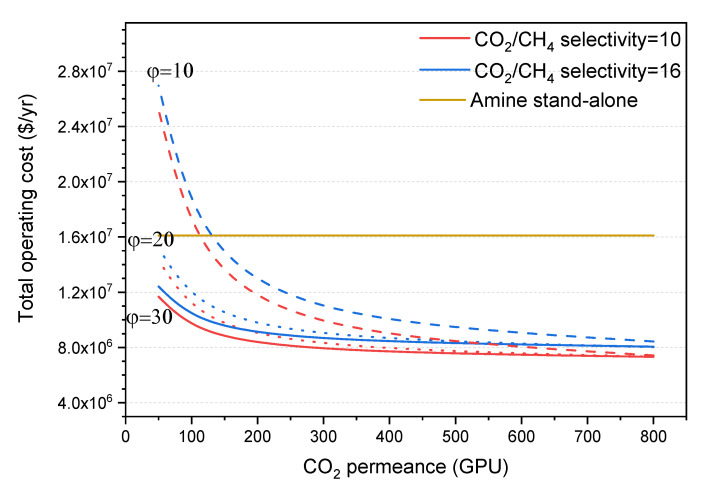
Effect of CO_2_ permeance on the total annual capital operating cost of the hybrid process at CO_2_/CH_4_ selectivity of 10 (red lines) and 16 (blue lines) and different pressure ratio values of 10 (dash lines), 20 (dot lines), and 30 (solid lines). The cost is compared to the operating cost of the stand-alone amine process (horizontal dark yellow line).

**Figure 6 membranes-10-00413-f006:**
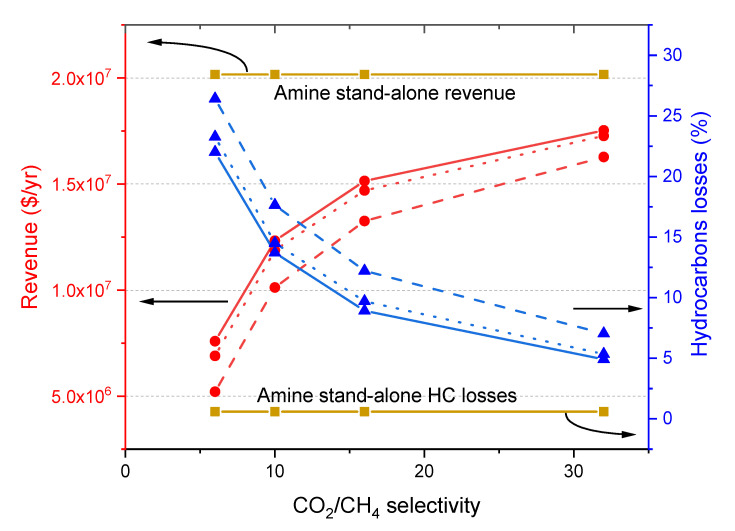
Effect of CO_2_/CH_4_ selectivity on revenues (left axis) and hydrocarbon losses (right axis) of the hybrid process at pressure ratio values of 10 (dash lines), 20 (dot lines), and 30 (solid lines). The values are compared to the stand-alone amine process (horizontal dark yellow line). Red curves correspond to revenue and blue curves correspond to hydrocarbon losses.

**Figure 7 membranes-10-00413-f007:**
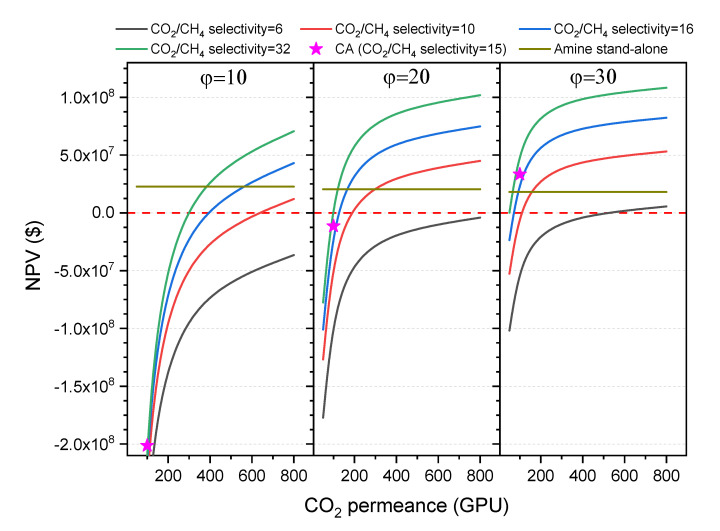
Effect of CO_2_ permeance on net present value (NPV) of the hybrid process at CO_2_/CH_4_ selectivity of 6 (black lines), 10 (red lines), 16 (blue lines) and 32 (green lines) and pressure ratio of 10 (left part), 20 (middle part), and 30 (right part). NPV values are compared to the value of the amine stand-alone process (horizontal dark yellow line) and hybrid process using the CA membrane (pink star symbol).

**Figure 8 membranes-10-00413-f008:**
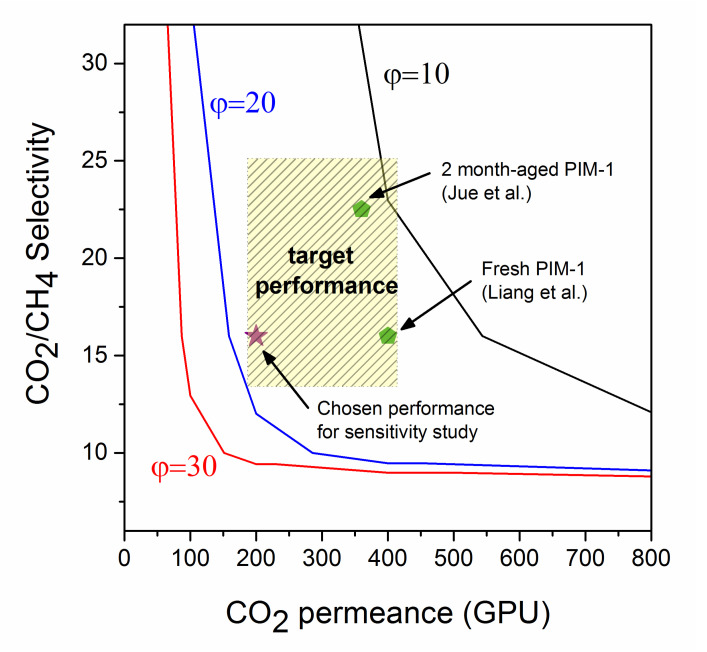
Summary of the feasibility of the hybrid process at various membrane performances (CO_2_ permeance and CO_2_/CH_4_ selectivity and three different studied pressure ratios of 10, 20, and 30 (represented as lines). Any points to the right-hand side of the lines are when the hybrid process becomes more feasible than the stand-alone amine process. Data points for PIM-1 membranes reported in the literature (fresh [37] and aged for 2 months [35]) are included).

**Figure 9 membranes-10-00413-f009:**
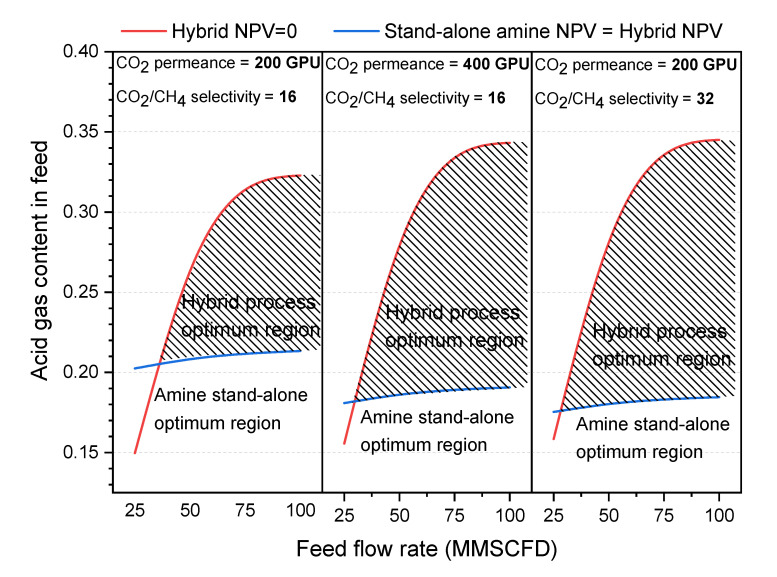
The hybrid process and the amine stand-alone process preferred ranges at various feed conditions (feed flowrate and acid gas content). Above the red lines, the hybrid process is not feasible (NPV <0), while below the blue lines, the amine stand-alone process is more economical (stand-alone amine NPV > hybrid process NPV).

**Figure 10 membranes-10-00413-f010:**
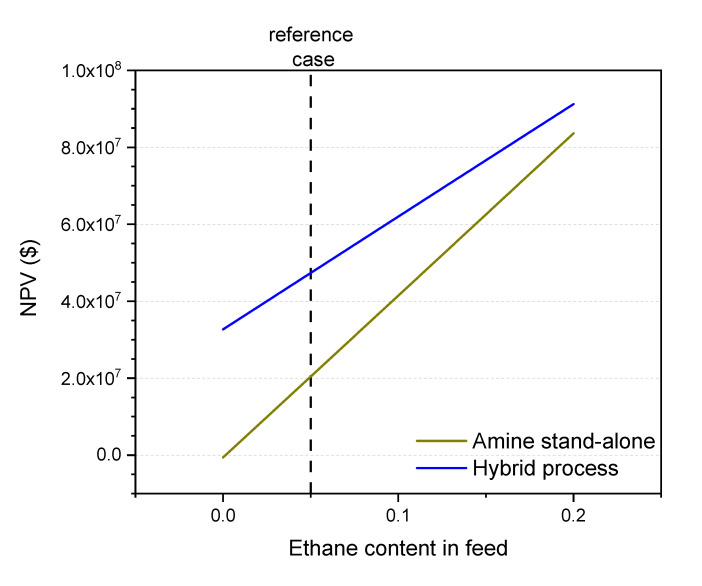
Effect of ethane content on the feasibility of the hybrid process (blue line) and the amine stand-alone process (dark yellow line). The reference case is shown.

**Figure 11 membranes-10-00413-f011:**
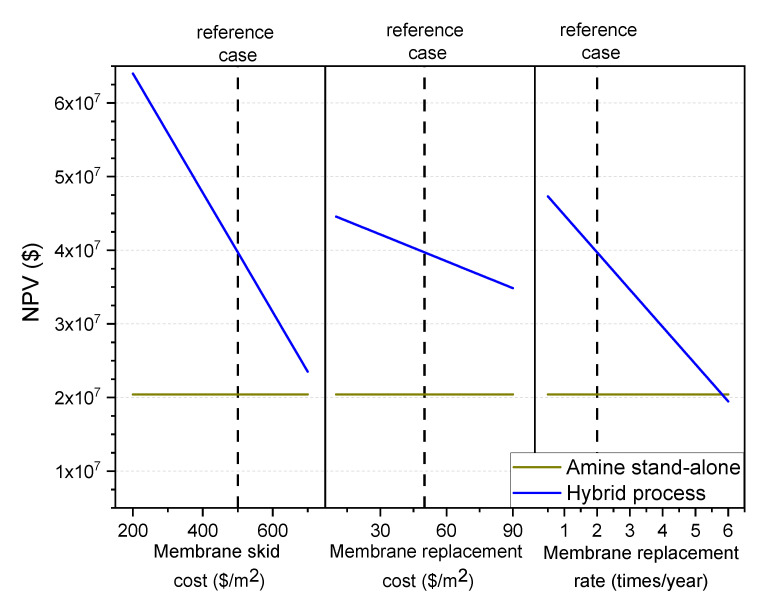
Effect of the membrane skid cost (left part), the membrane material cost (middle part), and the membrane replacement rate (right part) on the NPV value. The amine stand-alone process NPV is shown for comparison (dark yellow line).

**Table 1 membranes-10-00413-t001:** Summary of the recent performance of thin superglassy polymers. All the membranes were tested with pure gas.

Selective Layer	Gutter Layer	Support	Membrane Configuration and Thickness	Testing Pressure	CO_2_ Permeance [GPU]	N_2_ Permeance [GPU]	Estimated CH_4_ Permeance * [GPU]	CO_2_/CH_4_ Selectivity	Aging Time	Reference	Year
**PTMSP/PAF11**	-	PAN	Flat (kiss coating) 6.8 μm	2 bar	1900	280	350	5.4	450 days	[33]	2018
**PTMSP**	-	PAN	Flat (kiss coating) 2.8 μm	2 bar	310	19	24	13.1	600 days		
**PTMSP/PAF11**	-	PAN	Flat (kiss coating) 1.7 μm	2 bar	500	31	39	12.9	650 days		
**PIM-1**	-	PAN	Flat (dip coating) 0.7 μm	2 bar	3880 **	110 **	138	28.2	300 days	[34]	2015
**f-MWCNTs/PIM-1**	-	PAN	Flat (dip coating) 0.7 μm	2 bar	8200 **	220 **	275	29.8	300 days		
**PIM-1**	-		HF (spinning) 2.8 μm	100 psi	360	13	16	22.2	2 months	[35]	2017
**PIM-1**	-	PAN	Flat (coating)	2 atm	388	16	20	19.4	90 days	[36]	2018
**PIM-1/C-HCP (40 wt%)**	-	PAN	Flat (coating)	2 atm	9379	834	1043	9.0	90 days		
**PIM-1**	PDMS	PAN	HF (coating)	2 bar	402.6	18.9	24	17.0	Fresh	[37]	2018
**PIM-1 (4 wt% in CHCl_3_ and amylene)**	-	PAN	Flat (kiss coating) 6 μm	0.5–5 bar	144	3.7	5	31.1	Fresh	[38]	2019
**PIM-1 (2 wt% in CHCl_3_ and EtOH)**	-	PAN	Flat (kiss coating) 2.7 μm	0.5–5 bar	415.9	14.4	18	23.1	Fresh		
**PIM-1**	Cross-linked PTMSP	MFFK-1	Flat (kiss coating) 0.29–0.42 μm	3 bar	208–297	6 to 8	8 to 10	34 to 56	95 days	[39]	2019
**PIM-1**	PDMS/MOF	PAN	Flat (spinning)	2 bar	490	16	20	24.5	8 weeks	[32]	2020
**PIM-1/MOF-74-Ni**	PDMS/MOF	PAN	Flat (spinning)	2 bar	1200	40	50	24.0	8 weeks		
**PIM-1/NH2-UiO-66**	PDMS/MOF	PAN	Flat (spinning)	2 bar	900	35	44	20.6	8 weeks		

* Assumed based on N_2_ permeance (at 25% higher permeance than N_2_ gas). ** Converted from Nm^3^ m^−2^ h^−1^ bar^−1^ to GPU (1 Nm^3^ m^−2^ h^−1^ bar^−1^ = 340 GPU).

**Table 2 membranes-10-00413-t002:** Raw gas stream conditions.

**Feed Flow [MMSCFD]**	100	
**Temperature [°F]**	120	
**Feed pressure [psia]**	900/600/300	
**Permeate pressure [psia]**	30	
**Composition**	CH_4_	56.0%
	CO_2_	8.0%
	N_2_	10.0%
	H_2_S	20.0%
	H_2_O	0.1%
	C_2_H_6_	5.0%
	C_3_H_8_	0.9%

**Table 3 membranes-10-00413-t003:** The cost exponent factor m for different equipment in the amine gas plant [54].

Equipment	Cost Exponent Factor m
**Absorber**	1
**Regenerator**	1
**Heat exchangers**	0.8
**Pumps/Air cooler fans**	0.8
**Surge tanks**	0.6

**Table 4 membranes-10-00413-t004:** Direct and indirect cost factors and its percentage to purchased equipment.

	**Purchased Equipment**	**1**
	**Delivered cost factor**	**0.1**
	**Component**	Factor
**Direct cost factor**	Equipment erection	0.40
	Piping	0.65
	Instrumentation and control	0.30
	Electrical	0.25
	Process buildings and structures	0.45
	Service facility	0.75
	Site preparation	0.15
**Indirect cost factor**	Engineering/Supervision	0.33
	Construction expenses	0.40
	Legal expenses	0.10
	Contractors fee	0.10
	Contingency	0.40
	Total factor = Direct cost factor + Indirect cost factor + Delivered cost factor	5.38
**FCI** = Total factor × Equipment cost **WCI** = OPEX (first year) + Amine loading + start-up expenses **TCI** = FCI + WCI		

**Table 5 membranes-10-00413-t005:** Heating duty factor to amine circulation rate for the main equipment in the amine-based gas sweetening process [60].

[× Amine Circulation Rate (GPM)]	
**Reboiler [MMBtu/h]**	0.072
**Rich/lean amine heat exchanger [MMBtu/h]**	0.045
**Amine cooler [MMBtu/h]**	0.015
**Reflux condenser [MMBtu/h]**	0.03

**Table 6 membranes-10-00413-t006:** Gas sweetening power requirement factor based on the amine circulation rate [60].

[× Amine Circulation Rate (GPM)]	
**Main amine pump [hp]**	6.5 × 10^‒4^ × feed pressure (psig)
**Amine booster pumps [hp]**	0.06
**Reflux pumps [hp]**	0.06

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
