# Peer review of "Superglassy Polymers to Treat Natural Gas by Hybrid Membrane/Amine Processes: Can Fillers Help?"

_membranes, 2020, doi:10.3390/membranes10120413_

Round 1
Reviewer 1 Report
The manuscript presents a thorough economic comparison between a hybrid membrane process and a standalone amine process for natural gas sweetening. The study is surely interesting and useful to understand membrane potential in the field and to link that potential on actual membrane performance.
I think the manuscript can be accepted for publication once some minor issue are addressed to improve the overall clarity and readability.
Some example is reported below:
Page 8 line 99: the CO2/CH4 selectivity is considered as fixed to 0.85 but it is not clear how this value was chosen, indeed in the introduction values reported for superglassy membranes were generally higher (H2S/CO2 selectivity between 4 and 5).
Page 8 Amine solvent types and selection: DGA choice is reported and explained on the base of general qualitative consideration. I wonder if there is any study about comparison of the efficiency of the different solvents in the amine gas sweetening which could be added as a reference to possibly further, quantitatively, support this choice.
Page 9 line 141: the cost of the skid costs is chosen and the well-known paper from Baker and Lokhandwala is reported as a reference. As this work is rather old, I was wondering if the cost of the membrane was somewhat actualized and in case on the base of what considerations.
Page 10 line 174: It is not clear to me why the Equation 2 is referred as permeability equation in the text. Also it would be nice to known the work it is taken from.
Page 12 line 109: The number of workers is assumed to be 10. Is this number the same for both option? Would the simplicity of membrane operation allow to have a reduction of workers on the hybrid process?
Page 12 Results and discussion: It is not clear to me what are the specification of the different hybrid process considered. As an example the purity of the retentate or the amount of acid gas removed held constant while analyzing the cost of membranes with different performances? or the separation carried on in the different modules was optimized to lower the cost of the hybrid system? In the latter case how was the optimization carried on? which variables were adjusted? I think this has to be explained clearly in order to completely understand all the results.
Page 19 lines 410-415: This sentence is kind of confusing. From the chart it is clear that increasing the acid gas content the membrane process becomes more feasible, but here it is said that at 18% acid gas content the hybrid process is better while above 19% the amine is better and can be made more profitable by doubling the CO2 permeance!? Please check if something is missing.
Page 19 lines Ethane content in the feed. Please clearly define the reference case (membrane properties and the pressure ratio used for this analysis) and how the ethane was added. Is it substituting the methane?
Page 20 figure 10. In the figure there is a red dotted line on the x axis that seems not to indicate nothing. I suggest to remove it to avoid confusion.
Finally, as general comment, I would suggest the author to comment in the introduction and/or in the conclusion on the real industrial interest of the process considered. The work focuses on high acid gas content and the sensitivity analysis clearly shows that the hybrid process efficiency is optimized in this condition. However natural gas has usually lower acid gas content, so that it is not clear to what extent this process could be applicable in real operation. It would be nice to give an estimation of the amount of natural gas reservoir with this type of acid gas concentration.
Author Response
First of all we would like to thank the Reviewers for all the comments and the time they have put into reviewing our work. We have answered all the queries (see below) and have made amendments, which we believe have improved the clarity and quality of the manuscript. All the changes have been marked to facilitate the revision. We would like to bring to the Reviewers’ and the Editor’s attention that the numbering of lines in the original submission started on page 6 of the manuscript. We have amended it for the new submission.
Reviewer 1
Comments and Suggestions for Authors
The manuscript presents a thorough economic comparison between a hybrid membrane process and a standalone amine process for natural gas sweetening. The study is surely interesting and useful to understand membrane potential in the field and to link that potential on actual membrane performance.
I think the manuscript can be accepted for publication once some minor issue are addressed to improve the overall clarity and readability.
Some example is reported below:
- Page 8 line 99: the CO2/CH4 selectivity is considered as fixed to 0.85 but it is not clear how this value was chosen, indeed in the introduction values reported for superglassy membranes were generally higher (H2S/CO2 selectivity between 4 and 5).
Recently-reported amidoxime-functionalized ladder PIMs (e.g., AO-PIM-1) (DOI: 10.1126/sciadv.aaw5459) show a H2S/CO2 selectivity of about 5 due to penetrant-induced plasticization, however the H2S/CO2 selectivity for unmodified PIM-1 and other superglassy polymers is lower, for instance for PTMSP the selectivity has been reported to be as low as 1.2 (Acid Gas: Effect on Membrane Properties, Encyclopedia of Membranes). Therefore, we have selected that value (which corresponds to a CO2/H2S selectivity of 0.85) to make sure that we are in a more realistic scenario, keeping in mind that higher selectivities will be more favorable for the hybrid process. Also, we can expect laboratory-scale reported values in the literature to be lower at field condition.
- Page 8 Amine solvent types and selection: DGA choice is reported and explained on the base of general qualitative consideration. I wonder if there is any study about comparison of the efficiency of the different solvents in the amine gas sweetening which could be added as a reference to possibly further, quantitatively, support this choice.
We have included a new reference to a work where MEA and DGA solvents are compared. See below the text that has been added to the manuscript.
However, other aspects such as solvent degradation and solubility of heavier hydrocarbons need to be carefully considered when selecting the amine. It is recommended to use commercial software such as Aspen Plus to compare the performance of different solvents, as in the work by Bae et al. [51], where it was found that DGA led to significant energy savings as compared to MEA for natural gas containing high CO2 concentration.
- Page 9 line 141: the cost of the skid costs is chosen and the well-known paper from Baker and Lokhandwala is reported as a reference. As this work is rather old, I was wondering if the cost of the membrane was somewhat actualized and in case on the base of what considerations.
We agree with the reviewer, the value has been taken from a paper published twelve years ago, but unfortunately we have not been able to find a more recent value, and it is quite difficult to obtain the actual membrane skid cost from gas separation module suppliers. However, it is worth mentioning that we have carried out a sensitivity analysis on the skid cost that is presented in section 3.7.2, of the manuscript.
- Page 10 line 174: It is not clear to me why the Equation 2 is referred as permeability equation in the text. Also it would be nice to known the work it is taken from.
Sorry, this was a typo and “permeability” was meant to be “per” so we have amended it. We have added the reference.
- Page 12 line 109: The number of workers is assumed to be 10. Is this number the same for both option? Would the simplicity of membrane operation allow to have a reduction of workers on the hybrid process?
Yes, the reviewer is right, the simplicity of the membrane operation would allow the reduction of number of workers, but only if we were considering just a purely membrane process. Since this is a hybrid process and we still have the amine unit (even though it is a smaller size), we require the same number of workers.
- Page 12 Results and discussion: It is not clear to me what are the specification of the different hybrid process considered. As an example the purity of the retentate or the amount of acid gas removed held constant while analyzing the cost of membranes with different performances? or the separation carried on in the different modules was optimized to lower the cost of the hybrid system? In the latter case how was the optimization carried on? which variables were adjusted? I think this has to be explained clearly in order to completely understand all the results.
The membrane unit is designed to remove 60% of acid gases in all cases, with Aspen HYSYS calculating the required membrane area. We have indicated this at the beginning of the results section 3.1 to make it clearer. No process optimization was done in this work, but we mention in section 3.6 that it could be done by, for instance, finding the optimum percentage of acid gas removal, applying a two-stage process at very high feed acid gas content, or applying counter-current flow instead of crossflow, and that would lead to a wider favourable range than that presented in Figure 9. We have reworded this part in the manuscript to make it clearer.
- Page 19 lines 410-415: This sentence is kind of confusing. From the chart it is clear that increasing the acid gas content the membrane process becomes more feasible, but here it is said that at 18% acid gas content the hybrid process is better while above 19% the amine is better and can be made more profitable by doubling the CO2 permeance!? Please check if something is missing.
Thanks for suggesting more clarity to this point. We have modified the writing to make it clearer. Please find below the new text.
As can be seen in Figure 9, doubling the membrane performance increases the preferred hybrid process working region at both ends. However, doubling the CO2/CH4 selectivity seems to be slightly more attractive than doubling the CO2 permeance. For example, for an acid gas concentration of slightly above 18%, doubling the CO2/CH4 selectivity makes the hybrid process more favourable than the stand-alone one (right hand side graph in Figure 9), whereas doubling the permeance is not enough (see middle graph in Figure 9).
- Page 19 lines Ethane content in the feed. Please clearly define the reference case (membrane properties and the pressure ratio used for this analysis) and how the ethane was added. Is it substituting the methane?
As kindly requested by the Reviewer, we have added the reference membrane performance at the beginning of section 3.7, and have added in section 3.7.1 that in all cases the ethane content has been increased at the expense of reducing the amount of methane in the raw natural gas.
- Page 20 figure 10. In the figure there is a red dotted line on the x axis that seems not to indicate nothing. I suggest to remove it to avoid confusion.
The red dotted line has been removed.
- Finally, as general comment, I would suggest the author to comment in the introduction and/or in the conclusion on the real industrial interest of the process considered. The work focuses on high acid gas content and the sensitivity analysis clearly shows that the hybrid process efficiency is optimized in this condition. However natural gas has usually lower acid gas content, so that it is not clear to what extent this process could be applicable in real operation. It would be nice to give an estimation of the amount of natural gas reservoir with this type of acid gas concentration.
We have included in the introduction that due to the increasing demand in natural gas there is a need for exploiting reservoirs of lower quality gas, and have also included a reference to the International Energy Agency that includes some examples of fields with high acid gas content, such as in the Middle East (has large natural gas reservoirs that contain around 30% of H2S and 10% of CO2), and Russia’s Astrakhan field that contains 40% acid gases (around 25% H2S and 15% CO2)

Reviewer 2 Report
Interesting study, however its scope might be extended (i.e. more comparison with the currently available materials)
Lines 6-21
They conclude that "In normal glassy membranes, CO2 tends to permeate faster than H2S as the diffusion selectivity dominates the separation, leading to relatively low H2S/CH4 separation and thus hindering the use of glassy membranes to treat natural gas with high H2S concentrations [39]."
This conclusion needs a bit more discussion since there are many factors playing roles such as feed gas composition, total/partial pressure, other impurities and plasticisation.
Line 28
The cost is given in $/m3, is that correct or is it $/m2 ??
Section 2.1
(Line 93) "it was found that the hybrid process becomes more feasible if the membrane in the first stage removes around 45% of acid gas"
It is also stated in the report (Line 416-421) that "The optimisation process can involve finding out the optimum percentage of acid gas removal, applying a two-stage process at very high feed acid gas content, ....."
Are all calculations are based on this (45% of acid gas removal) or the authors also investigated this in this study and decided to use this value as the base for the calculations? If it was studied, I am not sure that the results are discussed and given in the report.
Section 2.2
The authors chose the membrane performance for their case study based on the data given in Table.1. It would be much better to visualize the data (in a graph) to make their choice more clear and understandable for the readers. This graph may also include the available membrane materials fort he application.
Section 3
Membrane area: It is mentioned that the calculation was performed by ASPEN HYSYS. Here it is stated that it could be calculated by the equation [???????? ????=?×(1+??2 ?????????)?]. Is this the model that was used for the calculation or is this the conclusion from the graph?
Section 3.6
Are these calculations based on the same removal ratio (45% of acid gas)?
Author Response
First of all we would like to thank the Reviewers for all the comments and the time they have put into reviewing our work. We have answered all the queries (see below) and have made amendments, which we believe have improved the clarity and quality of the manuscript. All the changes have been marked to facilitate the revision. We would like to bring to the Reviewers’ and the Editor’s attention that the numbering of lines in the original submission started on page 6 of the manuscript. We have amended it for the new submission.
Reviewer 2
Comments and Suggestions for Authors
Interesting study, however its scope might be extended (i.e. more comparison with the currently available materials)
- Lines 6-21
They conclude that "In normal glassy membranes, CO2 tends to permeate faster than H2S as the diffusion selectivity dominates the separation, leading to relatively low H2S/CH4 separation and thus hindering the use of glassy membranes to treat natural gas with high H2S concentrations [39]."
This conclusion needs a bit more discussion since there are many factors playing roles such as feed gas composition, total/partial pressure, other impurities and plasticisation.
Thanks for the suggestion. We have added some references about the partial pressure/ plasticization effects on glassy membranes. See below the added text.
However, the glassy membrane performance can be altered when applied at high acid gas partial pressures. For example, the H2S permeance might slightly exceed the CO2 permeance due to plasticization and sorption site competition [41-43].
- Line 28
The cost is given in $/m3, is that correct or is it $/m2 ??
Sorry, this was a typo. We have amended it and it now reads $/m2.
- Section 2.1
(Line 93) "it was found that the hybrid process becomes more feasible if the membrane in the first stage removes around 45% of acid gas"
It is also stated in the report (Line 416-421) that "The optimisation process can involve finding out the optimum percentage of acid gas removal, applying a two-stage process at very high feed acid gas content, ....."
Are all calculations are based on this (45% of acid gas removal) or the authors also investigated this in this study and decided to use this value as the base for the calculations? If it was studied, I am not sure that the results are discussed and given in the report.
Sorry for the confusion. All the simulations were done for a fixed acid gas removal value of 60% acid gas removal, without any further optimization. Therefore, we have removed the sentence in section 2.1 where it was indicated a 45% removal to avoid confusion. The optimum 60% value was obtained for one case (CO2 permeance of 200 GPU and CO2/CH4 of 16) at a pressure ratio of 30 (see graph below), but it was assumed the same for all cases, as at lower permeance and pressure ratios it only has a slight effect on the NPV.
An acid gas removal value of 60% in the membrane unit was found to be the most profitable target for a membrane with a CO2 permeance of 200 GPU and a CO2/CH4 selectivity of 16, and this value was fixed for all the simulations, as it had little effect on the selected economic indicators.
- Section 2.2
The authors chose the membrane performance for their case study based on the data given in Table.1. It would be much better to visualize the data (in a graph) to make their choice more clear and understandable for the readers. This graph may also include the available membrane materials for the application.
Based on reported values in the literature for TFC membranes (Table 1) we have assumed a performance range for the superglassy membrane of CO2 permeance (50 - 800 GPU) and a CO2/CH4 selectivities between 6 and 32. We think that plotting all these values with fresh and aged performance in a single graph would be a bit confusing. However, in Fig. 8 in the manuscript two points corresponding to reliable data in the literature for a fresh and an aged PIM-1 membranes, alongside a point corresponding to the performance selected for our sensitivity study, have been included.
- Section 3
Membrane area: It is mentioned that the calculation was performed by ASPEN HYSYS. Here it is stated that it could be calculated by the equation [???????? ????=?×(1+??2 ?????????)?]. Is this the model that was used for the calculation or is this the conclusion from the graph?
Membrane areas in all cases were determined by Aspen HYSYS, as indicated in section 2.4.2. We have indicated this again in section 3.1 for clarity.
- Section 3.6
Are these calculations based on the same removal ratio (45% of acid gas)?
As mentioned above, all the calculations are for a fixed value of 60% acid gas removal.

Reviewer 3 Report
The authors must address or comment on the following:
- Page 3, last line: Why is the performance going to drop in higher pressures? This is known to the membrane community but needs to be addressed in this publication.
- The authors need to mention the kinetic diameter of all gases tested or mentioned in their research. Currently only CO2 and H2S are mentioned. The same applied for the diffusion coefficient.
- What is the configuration of the membrane systems, is it in parallel or in series?
- Did the authors conduct a study on the return of investment (ROI)? This is needed for a feasibility study to be complete.
- Did the authors take into consideration the fouling effect in low temperatures?
- How does the higher pressure ratio affect the integrity of the membranes? This could be a factor that potentially increases the operating costs in page 14, line 298 rather than decrease them.
- What is the stability of the glassy membranes in H2S. H2S could corrode the glassy components and/or sealing. That is a common phenomenon, and it has to be incorporated in maintenance costs.
Author Response
First of all we would like to thank the Reviewers for all the comments and the time they have put into reviewing our work. We have answered all the queries (see below) and have made amendments, which we believe have improved the clarity and quality of the manuscript. All the changes have been marked to facilitate the revision. We would like to bring to the Reviewers’ and the Editor’s attention that the numbering of lines in the original submission started on page 6 of the manuscript. We have amended it for the new submission.
Reviewer 3
Comments and Suggestions for Authors
The authors must address or comment on the following:
- Page 3, last line: Why is the performance going to drop in higher pressures? This is known to the membrane community but needs to be addressed in this publication.
Thanks for the suggestion. We have added that this is due to plasticization.
…the membrane performance is expected to drop when tested at higher pressures with gas containing hydrocarbons and acid gases due to plasticization phenomena.
- The authors need to mention the kinetic diameter of all gases tested or mentioned in their research. Currently only CO2 and H2S are mentioned. The same applied for the diffusion coefficient.
We have added the kinetic diameters and the critical temperatures for all the gases in section 2.2.
- What is the configuration of the membrane systems, is it in parallel or in series?
The configuration of the membrane system consists of a single permeation stage in series, we have assumed a single membrane module for the simulations, but this will need to be split into several smaller parallel modules in a real plant depending on the feed flow rate and module size.
- Did the authors conduct a study on the return of investment (ROI)? This is needed for a feasibility study to be complete.
The ROI is a good economic indicator to study the feasibility of a project. However, unlike NPV, it doesn’t consider the time value of money. That is the reason why we have selected NPV to compare the various processes. But we have included below a graph showing the ROI for all processes, and the graph with NPV that is in the manuscript, both for a pressure ratio of 20. It can be seen that the same conclusions can be drawn as the profit is fixed during the whole period of 20 years.
- Did the authors take into consideration the fouling effect in low temperatures?
No, we didn’t, as this was out of the scope of the paper, but it will certainly have an impact on the performance of the process. We have only focused on the performance at standard temperatures that are those often used at laboratory-scale testing.
- How does the higher pressure ratio affect the integrity of the membranes? This could be a factor that potentially increases the operating costs in page 14, line 298 rather than decrease them.
Yes, higher pressure ratios can affect the integrity of membrane materials, but we haven’t considered that in our study. We have assumed that superglassy polymers such as PIM-1 can withstand pressures up to 60 bar without significant degradation. For instance in the work by Pinnau and Koros ((DOI: 10.1126/sciadv.aaw5459) they tested membranes up to 80 bar, and demonstrated long-term stability of the membranes up to 10 days at a pressure of 8.6 bar. However, as we mention in our manuscript in the conclusions section, there is an evident lack of long term performance and lack of testing at field condition that should be addressed in future research. We have added the following statement into section 3.3:
Regarding the pressure ratio, a larger value leads to lower operating costs, assuming long-term stability of the membranes.
- What is the stability of the glassy membranes in H2S. H2S could corrode the glassy components and/or sealing. That is a common phenomenon, and it has to be incorporated in maintenance costs.
In our study, we have assumed that the membranes would be replaced two times each year, which is anyway higher than usual. For example, Stern in his study Upgrading natural gas via membrane separation processes (Table 3) assumes that membranes are replaced once every four years even though H2S is present in the natural gas. In our study the membrane maintenance cost includes the replacement cost, which could be because of physical aging/degradation of the polymer, or damage due to harsh conditions.
